# *Myxococcus xanthus* as a Model Organism for Peptidoglycan Assembly and Bacterial Morphogenesis

**DOI:** 10.3390/microorganisms9050916

**Published:** 2021-04-24

**Authors:** Huan Zhang, Srutha Venkatesan, Beiyan Nan

**Affiliations:** Department of Biology, Texas A&M University, College Station, TX 77843, USA; hzhang@bio.tamu.edu (H.Z.); svenkatesan@bio.tamu.edu (S.V.)

**Keywords:** germination, bacterial cell wall, sporulation, germination, morphology

## Abstract

A fundamental question in biology is how cell shapes are genetically encoded and enzymatically generated. Prevalent shapes among walled bacteria include spheres and rods. These shapes are chiefly determined by the peptidoglycan (PG) cell wall. Bacterial division results in two daughter cells, whose shapes are predetermined by the mother. This makes it difficult to explore the origin of cell shapes in healthy bacteria. In this review, we argue that the Gram-negative bacterium *Myxococcus xanthus* is an ideal model for understanding PG assembly and bacterial morphogenesis, because it forms rods and spheres at different life stages. Rod-shaped vegetative cells of *M. xanthus* can thoroughly degrade their PG and form spherical spores. As these spores germinate, cells rebuild their PG and reestablish rod shape without preexisting templates. Such a unique sphere-to-rod transition provides a rare opportunity to visualize de novo PG assembly and rod-like morphogenesis in a well-established model organism.

## 1. Introduction

While spheres may seem simple and physically preferable, cells are seldom spherical. Instead, most cells invest extra energy to establish and maintain non-spherical shapes through a process known as “morphogenesis”. How cells use molecules at the nanometer scale to establish a defined morphology at micrometer scale has become a fundamental question in biology. Because of their relative simplicity, bacteria are excellent models for studying how genes and proteins determine cell morphology. Rods are the simplest non-spherical shapes adopted by many bacteria. Phylogenic studies suggest that the common ancestor of bacteria was rod-shaped, and that rod-like shapes are advantageous for cell survival [1,2,3]. Thus, the switch from primeval spheres to rods may mark the origination of bacteria, and represents a landmark morphological transition in evolution. Understanding this switch will likely uncover fundamental mechanisms of morphogenesis.

The word “morphogenesis” (from the Greek words “morphê” and “genesis”) means “the beginning of shape”. However, despite seeing “the surprising and bewildering riot of shapes” in bacteria [3], true morphogenesis is seldom studied, because we can rarely see the beginning of shape. In most bacteria, rigid peptidoglycan (PG) structures largely determine cell shape. Disruption of PG usually results in the loss of defined cell shape and, eventually, cell death [4,5]. Thus, in order to maintain their shape during growth, bacterial cells must add new PG subunits into existing PG structures [6,7,8,9]. When cells divide, the shapes of the daughter cells are already predetermined by their mothers. The preexistence of PG has become a central challenge that impedes progress in understanding bacterial morphogenesis. 

One way to circumvent this dilemma is to investigate how cells establish non-spherical shapes from PG-deficient spheres. Some bacteria can shed PG, thus losing their original morphology to cope with environmental stresses, especially when attacked by host immune systems and antimicrobial agents [10,11]. Among these PG-deficient cells, spheroplasts and L-forms derived from rod-shaped cells are able to regenerate rod shapes [11,12]. Importantly, a stable L-form state can be induced in many bacteria by genetically inhibiting the synthesis of PG precursors or upregulating membrane biosynthesis [13,14,15]. L-form cells of both *Bacillus subtilis* and *Escherichia coli* can rebuild rods when their genetic defects are rescued, which provides potential vehicles for the study of the de novo generation of non-spherical shapes [16,17]. However, these PG-deficient cells usually take on an array of irregular shapes, especially during proliferation, which makes reproducible live-cell imaging technically challenging [16,17,18]. Moreover, spheroplasts and L-form cells take several generations to restore their original cell shapes, which implies the involvement of cell division in morphogenesis [12,17]. These temporal steps make it difficult to track morphological changes in single cells and to attribute them to a simple PG assembly system. Therefore, in order to better understand bacterial morphogenesis, it would be very helpful to find a system in which homogeneous spheres generate non-spherical shapes independent of both division and pre-existing templates. 

*Myxococcus xanthus* is a Gram-negative bacterium that has been studied extensively for its motility, multicellular development, and predatory behaviors. Vegetative *M. xanthus* cells are long rods (1 µm in diameter and 5–10 µm in length) that go through cell cycles similar to other rod-shaped model organisms, such as *E. coli* and *B. subtilis* [19,20]. In response to certain chemicals, individual *M. xanthus* cells can degrade their PG thoroughly and form spherical spores [21,22]. During germination, these spores restore vegetative morphology by assembling rod-shaped PG de novo. As PG is usually essential for bacterial survival, and its assembly systems are well conserved in most bacteria, including *M. xanthus* [23], *M. xanthus* provides unique opportunities to better understand the mechanisms of PG assembly and rod-like morphogenesis.

## 2. PG and Cell Shape

The entire PG layer is a continuous, mesh-like macromolecule of glycan strands that are crosslinked via short peptides [24]. PG surrounds the whole cell, provides major mechanical support against turgor pressure, and defines cell shape throughout the life cycles of most bacteria [25]. As PG is chemically unique, and usually essential for cell survival, the synthesis and turnover of PG have been predominant targets for antibacterial treatments [26].

Cells synthesize lipid II, the PG subunit carried by lipids, using conserved enzymes in the cytoplasm and the cell membrane. Lipid II is flipped into the periplasm, where it is assembled into the existing PG structure [9]. PG assembly during vegetative growth relies on two conserved polymerization systems: the Rod complex and class A penicillin-binding proteins (aPBPs). The central components of Rod complex are RodA, PBP2 and MreB. RodA, a SEDS (shape, elongation, division, and sporulation) family transglycosylase (TGase), catalyzes the formation of glycosidic bonds in the glycan strands. PBP2, a member of the class B penicillin-binding proteins (bPBPs) that has transpeptidase (TPase) activity, crosslinks the short peptides on adjacent glycan strands [23,27,28,29,30]. MreB, an actin-like cytoskeletal protein that is conserved in most rod-shaped bacteria, is proposed to form the scaffolds that orchestrate PG assembly [31]. Different from the Rod components, aPBPs have both TGase and TPase activity. While the Rod system is essential for rod-like morphology [31,32,33], the functions of aPBPs are still not fully understood [34]. A recent work suggests that aPBPs contribute to shape maintenance indirectly by repairing PG defects [35]. In general, the Rod complexes reduce cell diameter, whereas aPBPs increase it [36]. Hydrolases also play important roles in PG growth, by generating small openings in the existing PG layer to allow for the insertion of new subunits. Thus, PG is a dynamic structure under the coordinated control of its polymerases and hydrolases [37]. 

## 3. The Sporulation and Germination of *M. xanthus*

Many bacteria form spores in order to survive under unfavorable environmental conditions. The endospores formed by Firmicutes such as Bacilli and Clostridia have been subject to extensive studies. In these organisms, the morphological differentiation from rod-shaped vegetative cells to oval spores begins with an asymmetric division, resulting in the formation of a smaller cell—the forespore—and a larger mother cell. Eventually, the forespore is engulfed by, and becomes wholly contained within, the mother cell. The resulting endospores contain intact and, in many cases, thickened PG. In addition, because the cell poles in forespores are generated through division, the mature endospores are also likely to preserve the information of cell polarity [38,39,40].

Strikingly different from endospore-forming bacteria, cell division is not involved in the sporulation of *M. xanthus*. *M. xanthus* can form spores using two distinct mechanisms, both of which exhibit the transformation of entire rod-shaped vegetative cells into spheres. First, in response to starvation, large groups of vegetative cells can aggregate on solid surfaces and build fruiting bodies, which are filled up with spores. Such spores are difficult to study because in order to release individual spores, the coats of fruiting bodies have to be broken and removed via intensive sonication [20]. In the second mechanism, individual vegetative cells can form dispersed, spherical spores in response to various chemical signals, such as glycerol, dimethyl sulfoxide (DMSO), and agents that inhibit PG synthesis or disrupt PG—including β-lactams, D-cycloserine, fosfomycin, and lysozyme [41,42]. In contrast to fruiting bodies that require millions of cells, solid surfaces, and days to form, chemical-induced sporulation can occur at low cell density, in liquid media, and in a few hours. For example, adding 0.5–1 M glycerol into rich liquid media induces the transition of *M. xanthus* from rod-shaped vegetative cells to spherical spores in 1–3 h [41]. These “quick” spores used to be considered artificial. However, we believe that chemical-induced sporulation is a naturally occurring process through which dispersed cells form spores rapidly under particular environmental stresses. First, many signals that induce quick sporulation are also present in the natural habitats of *M. xanthus*. Second, chemical-induced spores show typical characteristics of starvation-induced spores, such as considerable resistance to heat, desiccation, UV irradiation, sonication, detergents, and enzymatic digestion [43].

Unlike endospores, glycerol-induced *M. xanthus* spores are PG-deficient. Using high-performance liquid chromatography (HPLC) and transmission electron microscopy (TEM), Bui et al. first reported that such spores contained no detectable muropeptides [21]. This conclusion was later confirmed using cryo-electron tomography (cryo-ET) [22]. Thus, during sporulation, vegetative cells thoroughly degrade their PG, shrink into near-perfect spheres [21], and synthesize spore coats that consist of polysaccharides and proteins [44]. Sensing certain environmental signals, such as inorganic ions HPO_4_^2−^, Mg^2+^, Ca^2+^, and NH_4_^+^, *M. xanthus* spores regenerate vegetative cells with rod-like morphology [45]. So far, Ca^2+^ has been found to be a strong germinant that induces roughly synchronized germination [46]. When incubated in rich liquid media containing Ca^2+^, *M. xanthus* spores can restore vegetative cell shape within 2–3 h [22,46,47].

## 4. PG Assembly and Morphological Transition during *M. xanthus* Spore Germination

Do *M. xanthus* spores preserve PG subunits from previous vegetative cells? Bui et al. proposed that glycerol-induced *M. xanthus* spores may contain PG subunits [21]. Consistent with this report, these spores are able to germinate and restore vegetative morphology in the presence of fosfomycin, an antibiotic that inhibits the production of UDP-MurNAc—a precursor of PG. However, after the exhaustion of preserved PG subunits, nascent vegetative cells become sensitive to fosfomycin, and are unable to elongate further [22,48]. These results indicate that glycerol-induced spores preserve most of the PG subunits from previous vegetative cells. Thus, PG assembly, rather than the production of PG subunits, is the decisive step for the restoration of rod shape. 

Similar to endospores [49], *M. xanthus* spores are refractile, and appear bright under a phase-contrast microscope [46,47]. Upon being suspended in rich media containing Ca^2+^, most spores lose their refractility within a few minutes, indicating that germination initiates immediately upon induction [46,47]. However, germinating spores remain spherical, and do not elongate until 45–60 min later [22]. Despite the absence of morphological changes, PG assembly initiates immediately once germination starts, as germinating spores begin to incorporate TAMRA 3-amino-D-alanine (TADA)—a fluorescent D-amino acid (FDAA)—evenly onto their surfaces (Figure 1A) [22]. Because FDAAs specifically label newly assembled PG [50,51], the incorporation pattern of TADA indicates that spores first synthesize spherical PG layers in this early phase of germination.

After remaining spherical for approximately one hour, germinating spores quickly start to elongate. Compared to vegetative cells, which double their length in about four hours, the elongation of germinating spores proceeds quite rapidly, growing one cell length within an hour [22]. During elongation, cells only incorporate TADA to the cylindrical, non-polar portion of their surfaces, indicating that the cell poles become inert for PG growth (Figure 1A) [22].

These distinct growth patterns allow us to divide *M. xanthus* spore germination into two phases: the spherical phase (Phase I), in the first hour of germination, when spores assemble PG evenly on their entire surfaces; and the elongation phase (Phase II), in the second to third hours, when PG growth at the non-polar regions drives cell elongation (Figure 1A). The correlation between the shapes of cells and the patterns of PG growth suggests that a major switch in the mode of PG assembly results in the de novo establishment of rod shape.

## 5. Roles of aPBPs and the Rod System during Germination

The roles aPBPs and the Rod system play in PG assembly can be studied by monitoring TADA incorporation in the presence of antibiotics, which specifically inhibit either aPBPs or the Rod system. In the spherical phase, neither mecillinam—an inhibitor of PBP2 in the Rod system—nor cefsulodin or cefmetazole—antibiotics that mainly inhibit aPBPs—are able to block TADA incorporation. However, spores stop incorporating TADA when treated with all three antibiotics. In contrast, once spores enter the elongation phase, mecillinam alone is sufficient to block TADA incorporation, whereas cefsulodin and cefmetazole do not show significant effects [22]. *M. xanthus* spores are able to germinate into rods in the presence of cefsulodin and cefmetazole, indicating that aPBPs are dispensable for cell elongation [22]. However, elongated cells revert to spheres after prolonged inhibition of aPBPs, which suggests that aPBPs stabilize rod shapes [22]. This observation echoes the reports on *E. coli,* where aPBPs did not determine rod shape, but rather maintained PG integrity, especially across different pH environments [35,52]. Whereas both aPBPs and the Rod system participate in the assembly of spherical PG layers, the Rod system is essential for the establishment of rod shape. 

The Rod proteins are stably produced, and remain active during the entire germination process [22,44], whereas the spatial distribution of the Rod complexes alters dramatically when the spores start to elongate. While RodA localizes randomly on spore surfaces in the spherical phase, it concentrates at non-polar regions during elongation [22]. Similarly to RodA, MreB first appears diffusive in the spherical phase, and then forms filaments that mainly localize at non-polar regions as nascent cells elongate [22]. Such non-polar localization of Rod complexes is consistent with the lateral growth patterns of PG. Taken together, to establish rods from spheres, germinating *M. xanthus* spores must first establish cell poles, and then restrict Rod complexes to non-polar regions.

## 6. De Novo Establishment of Cell Poles

While *M. xanthus* spores are approximately spherical, those spheres are rarely “perfect”. If such morphological “imperfection” preserves polarity from previous vegetative cells, each germinating spore is expected to elongate along the long axis of the ovoid. However, the elongation axis during germination appears random in many spores, independent of their original morphology [22].

Then how do spores establish cell poles de novo? In vegetative *M. xanthus* cells, directed motility requires a head–tail polarity axis, which is under the modulation of multiple regulators, including the Mgl regulators (MglA, MglB and MglC) [53,54,55], the RomR–RomX pair [56,57,58], and PlpA [59]. Among these regulators, MglA—a Ras-like GTPase—and its GTPase-activating protein—MglB—are required for rapid cell elongation in the elongation phase of germination. Specifically, fine-tuned MglA is critical for optimal germination efficiency. Deleting MglA or MglB, nullifying the active site of MglA, or overstimulating the GTPase activity of MglA by overproducing MglB all cause severe delays in cell elongation (Figure 1A) [22]. Strikingly, unlike the wild-type spores, which maintain relatively smooth cell surfaces throughout the germination process, these Mgl mutant spores germinate into bulged, multipolar intermediates (Figure 1A). Many of these intermediates lyse before becoming vegetative cells, especially under osmotic stress [22]. Such morphological abnormality directly reflects the structural defects in newly assembled PG. A large portion of Rod complexes heavily incorporate TDA, mislocalize at cell poles, and bulge in elongating Mgl mutant spores. This is in contrast to wild-type spores in which Rod complexes, and therefore PG assembly, are restricted to non-polar regions in the elongation phase of germination (Figure 1A) [22]. These findings indicate that the Mgl system plays a central role in expelling PG assembly from cell poles during germination.

## 7. Symmetry Breaking by Random Fluctuation

Does polarity originate from preserved spatial cues, or from stochastic fluctuations? During sporulation *M. xanthus* retains both MglA and MglB in its spores [22,60,61]. Here, fluorescence-labeled MglB is seen to form a single bright cluster in each spore throughout the entire course of germination. These clusters first move randomly in the spherical phase, and then abruptly stop moving. Importantly, once an MglB cluster stalls, the spore immediately starts to elongate, using the position of the MglB cluster as one cell pole (Figure 1B) [22]. In contrast, MglA only begins to form a cluster after MglB stalls, and the formation of MglA clusters requires MglB. Once formed, clusters of MglB and MglA always occupy opposite sides of the same spore (Figure 1B) [22]. Thus, the sequential stabilization of MglB and MglA clusters establishes the polarity axis for PG assembly. 

The random “walk” of MglB clusters during the spherical phase of germination suggests that polarity is not preserved. The localizations of stalled MglB clusters do not correlate with the geometry of the spores. Instead, MglB clusters are immobilized by the inhibitors of PG assembly—especially the agents that inhibit the Rod system, such as mecillinam [22]. Thus, MglB clusters could stall at the sites where PG assembly has been completed, or has not yet initiated. In this case, it is unlikely that PG assembly has not yet initiated, because no obvious PG assembly activity is observed near the nascent cell poles after the stall of MglB clusters (Figure 1A) [22]. MglB activates the GTPase activity of MglA, and turns MglA-GTP into MglA-GDP. As MglA-GDP cannot form clusters, clusters of MglB and MglA (MglA-GTP) always maintain the farthest distance possible in the same cell [53,54]. As a consequence, once an MglB cluster stalls at one pole, the expulsion between MglB and MglA-GTP causes MglA-GTP to cluster at the opposite side of the spore—the second cell pole (Figure 1B).

## 8. When PG Assembly Intersects with Gliding Motility

MglB clusters must have the ability to survey the random fluctuation of PG growth during the spherical phase of germination, and stall at the region where PG growth completes first. Once polarity is established, Mgl regulators must expel the Rod complexes from the cell poles. How do Mgl regulators connect to the Rod system? Besides being a component in the Rod system, MreB also supports the gliding motility of *M. xanthus*. MglA-GTP stimulates the assembly of the gliding machineries through direct interaction with MreB [62,63,64,65], and then directs them toward non-polar regions [66]. The gliding motors then carry MreB filaments as they move rapidly in the membrane [48,67,68]. MglA-GTP clusters therefore co-localize with MreB filaments that lso carry Rod complexes [53,54,63,69]; however, MglB clusters do not (Figure 1B). 

Through the mutual expulsion between MglB and MglA-GTP, MglB clusters stall at the sites where PG assembly is complete and Rod complexes are absent. At the first pole, which contains the MglB cluster, MglB expels MglA-GTP, and thus the Rod complexes, toward the second pole. MglA-GTP then occupies the second pole and stimulates the assembly of the gliding machineries [63,70], which transport the Rod complexes toward the first pole [48,66]. As a result, the diametrically opposing clusters of MglA-GTP and MglB restrict the Rod system, and thus the assembly of PG, to non-polar locations (Figure 1B).

## 9. Perspectives

### 9.1. What Does It Take to Make a Rod?

As the assembly of PG is widely conserved in bacteria, the mechanisms by which *M. xanthus* rebuilds rods from spheres might reveal the common principles for rod-like morphogenesis. During *M. xanthus* spore germination, the Rod system is the only element known to be essential for the establishment of rod shape. Similarly, to our knowledge, artificial spheres from other rod-shaped cells are not able to restore their original shapes in the absence of the Rod system [71]. Given their critical roles, we hypothesize that restricting Rod complexes to a non-polar, but expanded region, might be a common prerequisite for building and maintaining a rod. On the one hand, when this restriction is relieved in the Mgl mutants, the cells generate poles randomly and display bulged morphology. On the other hand, in bacteria that over-restrict Rod complexes to very narrow regions, the cells are naturally oval [72,73]. 

### 9.2. The Versatile MreB

In comparison to its analogs in other bacteria, MreB in *M. xanthus* is truly extraordinary for its connection to the Mgl regulators and gliding machineries. MreB filaments in *M. xanthus* display a rapid, directed motion that is not yet reported in other organisms [48]. On the other hand, as in most rod-shaped bacteria, *M. xanthus* MreB plays conserved roles in PG assembly. As MreB filaments are intrinsically curved, and bind to the cytoplasmic membranes, the balance between filament bending and membrane deformation can lead MreB filaments to localize at inwardly curved regions [31,74,75,76,77,78]. This localization preference, in turn, affects the localization and dynamics of Rod complexes, which could be sufficient for the maintenance of rod shape [31,32,33,71,76,79,80]. Additionally, the composition and fluidity of the cytoplasmic membrane could also modulate the localization and dynamics of MreB [81,82,83,84]. For instance, rafts of anionic phospholipids preferentially enrich MreB monomers at the cell poles, and expel MreB filaments to non-polar regions [81]. Using these mechanisms, cells are able to generate rod shape spontaneously, based on random fluctuations, albeit much slower. For example, the Mgl and motor mutant spores can still elongate into rods and correct morphological defects after prolonged germination, and the artificial spheres of *E. coli* and *B. subtilis* can regain rod shapes after several generations [17,22]. In addition, Mgl and gliding motors are dispensable for vegetative growth [67,85]. Thus, rather than being the determinants of symmetry breaking, Mgl and gliding motors are supplementary accelerators. Nevertheless, such accelerators provide critical advantages for the survival of *M. xanthus* spores. As chemical-induced *M. xanthus* spores are dispersed and PG deficient, they are vulnerable to biotic and abiotic environments during germination. Equipped with the Mgl regulators and gliding machineries, *M. xanthus* spores are able to regain fitness within one generation. 

While the Rod complexes move relatively slowly, with nm/s velocities, gliding motors in *M. xanthus* move significantly faster, at µm/s [48,86,87,88]. Then how do MreB filaments co-ordinate multiple functions that require distinct dynamics? Aside from PG assembly, MreB affects a broad range of cellular functions, either directly or indirectly—such as membrane organization, DNA replication and segregation, twitching motility, and pathogenesis [89,90,91,92,93,94,95,96]. Studying MreB in *M. xanthus* provides an opportunity to understand how MreB organizes multiple functions simultaneously.

### 9.3. Are Small GTPases the Universal Regulators of Cell Polarity?

MglA-like GTPases distribute widely in phylogenically diverse bacteria. MglA and its eukaryotic homologs are proposed to have evolved from a common ancestor [97]. GTPase-mediated cell polarization is common in eukaryotes. The Rho-family GTPase Cdc42 and its homologs widely exist, in organisms ranging from yeast to human beings [98]. The rod-shaped yeast *Schizosaccharomyces pombe* forms spherical spores. During germination, Cdc42 first moves randomly during the isotropic growth phase, before stalling at a future pole [98,99]. Analogous to the connection between MglA and the Rod system in *M. xanthus*, Cdc42 in *S. pombe* is able to survey the integrity of the spore cell wall by interacting with cytoskeletons, cell wall-related enzymes, and molecular motors [99]. Such striking similarities suggest that *M. xanthus* might preserve a prototype of a polarity regulation system that evolved before the divergence between prokaryotes and eukaryotes. Therefore, studying the interactions between Mgl regulators, MreB, and motor-associated proteins may also aid in the understanding of cell polarity in a broad range of organisms.

### 9.4. What Are the Primary Roles of the Gliding Motors?

Vegetative *M. xanthus* cells move on surfaces using both gliding and twitching motility. In contrast to twitching—which exists in many phylogenically diverse bacteria—gliding, driven by fluid motor complexes, is rather unique to Myxococcales [100,101]. While twitching is required for group behaviors such as coordinated migration, predation, and multicellular development—to name but a few—gliding is rather secondary for vegetative growth [19,102]. Besides accelerating germination, the gliding machineries are also the distributors of spore coat polysaccharides during sporulation, which is critical for the resilience of the spores [22,103]. Thus, rather than facilitating cell locomotion, the primary function of the gliding machineries might be the distribution of various protein complexes in the cell membranes. 

### 9.5. How Do PG Polymerases Co-Ordinate with Hydrolases?

It is commonly accepted that the insertion of new PG subunits is associated with the local hydrolysis of the existing PG network [9,25,37]. It is therefore reasonable to hypothesize that a regulated co-ordination exists between PG polymerases and hydrolases. Hence, exploiting the synergy between these two types of enzymes could usher in new treatments for bacterial infections [104,105]. As of now, our understanding of PG hydrolases has long been hampered by several challenges. First, these enzymes are highly redundant in most bacteria, where strains lacking single hydrolases usually do not show significant growth defects. Second, as the uncontrolled action of PG hydrolases potentially leads to cell lysis, it is difficult to observe highly activated PG hydrolysis during normal cell growth [37]. The sporulation process, in which vegetative *M. xanthus* cells degrade their PG thoroughly within two hours [21,22,41,106], sets the perfect stage for the study of PG hydrolases. In order to facilitate rapid PG degradation, the balance between PG polymerases and hydrolases changes, and hydrolysis becomes dominant over synthesis. Thus, studying PG hydrolases during sporulation could provide valuable insight into how PG polymerase co-ordinates with hydrolase. 

In conclusion, the complete degradation of PG during chemical-induced sporulation makes *M. xanthus* an invaluable model organism for investigating the dynamics of PG and cell morphology. First, when *M. xanthus* spores germinate, cells must rebuild their walls and re-establish rod shape without pre-existing PG as a template, matching the definition of “morphogenesis” perfectly. Second, unlike spheroplasts and L-forms, nascent *M. xanthus* cells restore rod-shape within one generation [22], which largely excludes the involvement of cell division. Third, the germination progress of individual *M. xanthus* spores can be tracked using simple bright-field imaging techniques, such as phase-contrast and differential interference contrast microscopy. At the population level, germination progress can be easily quantified using the aspect ratio (length/width) of individual spores [22]. Using *M. xanthus* as a model to study PG dynamics and cell morphology may allow us to answer many questions regarding bacterial growth and survival. 

## Figures and Tables

**Figure 1 microorganisms-09-00916-f001:**
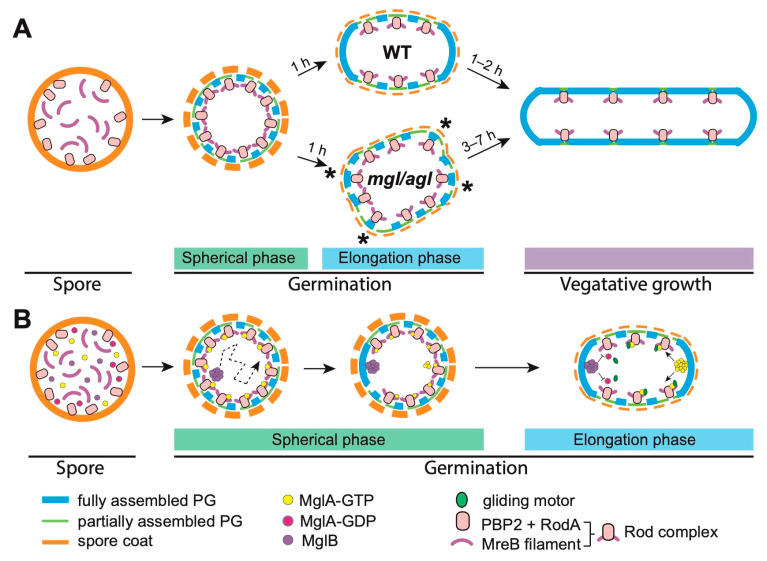
A schematic model for the de novo establishment of rod-shape from spherical, PG-deficient spores. (**A**) As the wild-type (WT) spores germinate, nascent cells restrict Rod complexes, and thus PG assembly, to non-polar regions in the elongation phase of germination. Such a pattern of PG growth maintains the integrity of cell surfaces and restores vegetative morphology within 3 h. In contrast, in the *mgl* spores (from the strains of *∆mglA*, *∆mglB*, and *mglA^Q82L^* that encode a GTPase-inactive variant of MglA, and *mglB^OE^* that overexpresses MglB) and the *agl* spores that express truncated gliding motors, the localizations of Rod complexes and PG assembly are not restricted. As a result, such spores grow into bulged intermediates that are sensitive to osmotic stresses and restore vegetative cell shape much more slowly (4–7 h). The asterisks mark the sites where Rod complexes mislocalize and bulges appear. (**B**) Symmetry breaking by MglB and MglA. The GTP-bound form of MglA (MglA-GTP) connects to Rod complexes via MreB filaments. Through the mutual expulsion between MglB and MglA-GTP, MglB clusters survey the status of PG synthesis indirectly, and cannot localize at the sites where PG assembly is active and Rod complexes are present. Therefore, MglB clusters move randomly in the early spherical phase, when Rod complexes distribute randomly on cell surfaces. Once a patch of PG is completely assembled and the Rod complexes leave, the MglB cluster will stall at this site, which will become the first future pole. At the first pole, MglB expels MglA-GTP, and thus the Rod complexes, toward the second pole. MglA-GTP then occupies the second pole, stimulating the assembly of the gliding machineries, which transport MreB filaments, together with Rod complexes, toward the first pole. As a result, the diametrically opposing clusters of MglA-GTP and MglB restrict the Rod system, and thus the assembly of PG, to non-polar locations.

## Data Availability

Not applicable.

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
