# Peer review of "Myxococcus xanthus as a Model Organism for Peptidoglycan Assembly and Bacterial Morphogenesis"

_microorganisms, 2021, doi:10.3390/microorganisms9050916_

Round 1

Reviewer 1 Report

It is difficult to read this review article. Because the article title differs from the content and the content is limited and fragmented. The authors describe Myxococcus xanthus rather than bacteria in general. I cannot understand why M. xanthus is a model organism for bacterial cell morphogenesis. The authors should describe M. xanthus more in detail in Introduction. 

In the original article of PNAS, the authors used the words "phase 1" and "phase 2". But, in this review article, they used "spherical phase" and "vegetative growth". Does the spherical phase mean spore? If so, in this review article, morphological change means germination of spores. Thus, in the spherical phase, it is uncertain whether the spores can divide or not. Are spherical cells found in the vegetative growth? The authors explained the difference between spores of M. xanthus and endospores of Firmicutes in the section "The sporulation and germination of M. xanthus". I cannot understand why the "spores" of M. xanthus are recognized as spores. The authors should describe what are spores. 

In the M. xanthus spore germination, Mgl proteins play an important role. The authors mentioned that Mgl of M. xanthus may related to Cdc42 of the eukaryote Schizosaccharomyces pombe. The explanation for Mgl and Cdc42 is inadequate and fragmented. The authors should explain the Mgl proteins more detail in this review article. For example, how Mgl proteins are distributed among Bacteria? Are Mgl proteins related to only Myxococcus? or spore forming bacteria? or Gram-negative bacteria? 

The authors did not show the cell cycle of M. xanthus. They showed only gemination of M. xanthus spores. In order to show that M. xanthus is a model bacterium for bacterial cell morphogenesis. The authors should show the cell cycle and explain when DNA is replicated, and cell wall and membrane are synthesized. 

Author Response

It is difficult to read this review article. Because the article title differs from the content and the content is limited and fragmented. The authors describe Myxococcus xanthus rather than bacteria in general. I cannot understand why M. xanthus is a model organism for bacterial cell morphogenesis. The authors should describe M. xanthus more in detail in Introduction. 

We added "As PG is essential for bacterial survival and its assembly systems are well conserved in most bacteria, including M. xanthus [23], M. xanthus provides unique opportunities to understand the mechanisms of PG assembly and rod-like morphogenesis. "

In the original article of PNAS, the authors used the words "phase 1" and "phase 2". But, in this review article, they used "spherical phase" and "vegetative growth". Does the spherical phase mean spore? If so, in this review article, morphological change means germination of spores. Thus, in the spherical phase, it is uncertain whether the spores can divide or not. Are spherical cells found in the vegetative growth? The authors explained the difference between spores of M. xanthus and endospores of Firmicutes in the section "The sporulation and germination of M. xanthus". I cannot understand why the "spores" of M. xanthus are recognized as spores. The authors should describe what are spores. 

We believe that we have already explained the different phases in germination in our original manuscript. We wrote "These distinct growth patterns allow us to divide M. xanthus spore germination into two phases: the spherical phase (Phase I) in the first hour of germination when spores assemble PG evenly on their entire surfaces and the elongation phase (Phase II) in the second to third hour when PG growth at the nonpolar regions drives cell elongation (Fig. 1A). " To further distinguish spore and the spherical phase of germination, we also modified Fig. 1.

In the M. xanthus spore germination, Mgl proteins play an important role. The authors mentioned that Mgl of M. xanthus may related to Cdc42 of the eukaryote Schizosaccharomyces pombe. The explanation for Mgl and Cdc42 is inadequate and fragmented. The authors should explain the Mgl proteins more detail in this review article. For example, how Mgl proteins are distributed among Bacteria? Are Mgl proteins related to only Myxococcus? or spore forming bacteria? or Gram-negative bacteria? 

We added "MglA-like GTPases distribute widely in phylogenically diverse bacteria. MglA and its eukaryotic homologs are proposed to have evolved from a common ancestor [96]. "

The authors did not show the cell cycle of M. xanthus. They showed only gemination of M. xanthus spores. In order to show that M. xanthus is a model bacterium for bacterial cell morphogenesis. The authors should show the cell cycle and explain when DNA is replicated, and cell wall and membrane are synthesized. 

We clarified that "Vegetative M. xanthus cells are long rods (1 mm in diameter and 5 – 10 mm in length) that go through cell cycles similar to other rod-shaped model organisms, such as E. coli and B. subtilis [19,20]."

Reviewer 2 Report

This is a very nice overall view of both the question of morphogenesis and the use of M. xanthus in defining how one species overcomes a variety of spacial and temporal challenges to go from rod to sphere to rod.  The broad background that is used to validate the work and hypothesis.  This is a very nice review.

Author Response

We appreciate the reviewer's support.

Reviewer 3 Report

The manuscript by Zhang et al reviews how peptidoglycan (PG) synthesis during spore germination directs de novo formation of rod shaped cells in M. xanthus. Specifically, spherical spores, which contain no PG, transition into a rod shape because the location of PG synthesis is controlled. Here, a nascent pole is determined by MglB clusters, which in turn excludes MglA-GTP and triggers it to be nucleated at the opposite side of the spore, thus creating the future second pole. Once PG patches have been completed at the nascent poles, future PG synthesis only occurs at nonpolar regions resulting in a rod shaped cell (Fig. 1). The review highlights how M. xanthus serves as a unique and important model for addressing fundamental questions about bacterial morphogenesis. Other systems that address similar questions rely on forming artificial spheroplasts that take several generations, and thus cell divisions, to reestablish a rod. This review is timely as it discusses a hot topic in microbiology and it focuses on very recent findings from the Nan group. The review is rich in mechanistic details and is well organized. Below are specific suggestions to improve clarity.

Comments/suggestions:

In Figure 1A it might be helpful to show a spore before germination. The legend/figure show the “spherical phase” has “uncompleted PG” and “completed PG.” This is a bit confusing as the text says a spore, before germination, has no PG and only has PG subunits. Additionally, in this regard, is there a difference between PG subunits and uncompleted PG?

During vegetative growth does the gliding motility machinery, e.g. MglA, MglB, et al, play a role in PG synthesis? This should be commented on.

There was some redundancy in the text, including the figure, which could be streamlined.

Minor suggestions:

Line 30, this is a grandiose statement that is speculative that needs to be qualified.  Thus insert “may” before “mark…” or similar qualifier.

Line 41. Suggestion, change “become a central challenge and progress in understanding” to “become a central challenge that impedes progress in understanding”

Line 44. Insert “thus” before “losing”

Line 48. Replace “Especially, stable…” with “Importantly, the stable…”

Line 53. Would this sentence be clearer by inserting “reproducible” before “live-cell imaging…”

Line 56. Change “It becomes difficult…” to “These temporal steps make it difficult…”

Line 57. Insert “better” before “understand”

Line 58. Replace “is critical” with “would be very useful”

Line 66. Replace “angles” with “opportunities”

Line 105. After “First,…” insert “in response to starvation…”

Line 109. Change “Second” to “In the second mechanism,…”

Line 113. Replace “complete” with “occur”

Line 118. Insert “particular” after “under”

Line 119. Replace “could also” with “are”

Line 120. Replace “genuine” with “starvation-induced”

Line 151. Change to “labels”

Lines 235-36. Sentence a bit awkward. How about “During development M. xanthus retains the MglA and MglB protein within their spores. Here, fluorescently-labeled MglB is seen to form …”

Line 247. Not clear what “inhibitors” refers to. Are they known?

Line 253-54. Insert “are” before “always” and replace “longest” with “the farthest distance possible…”

Line 254. Change “As the consequence” to “As a consequence”

Line 270. Insert “move” after “complexes”

Line 286: Insert “a” after “display”

Line 303: Replace “and the” with “, similar to how”

Author Response

The author appreciate the reviewer's constructive comments. We have made changed accordingly. Please see below.

Comments/suggestions:

In Figure 1A it might be helpful to show a spore before germination. The legend/figure show the “spherical phase” has “uncompleted PG” and “completed PG.” This is a bit confusing as the text says a spore, before germination, has no PG and only has PG subunits. Additionally, in this regard, is there a difference between PG subunits and uncompleted PG?

We thank the reviewer for this suggestion. In the updated Fig.1, we show spore so that the dormant spores are differentiated from the spherical phase of germination. We changed "completed PG" and "uncompleted PG" to "fully assembled PG" and "partially assembled PG", respectively. "PG subunits" are the chemical units, UDP-MurNAc, which is mentioned in the text. 

During vegetative growth does the gliding motility machinery, e.g. MglA, MglB, et al, play a role in PG synthesis? This should be commented on.

There was some redundancy in the text, including the figure, which could be streamlined.

Minor suggestions:

Line 30, this is a grandiose statement that is speculative that needs to be qualified.  Thus insert “may” before “mark…” or similar qualifier.

Corrected.

Line 41. Suggestion, change “become a central challenge and progress in understanding” to “become a central challenge that impedes progress in understanding”

Corrected.

Line 44. Insert “thus” before “losing”

Added.

Line 48. Replace “Especially, stable…” with “Importantly, the stable…”

Corrected.

Line 53. Would this sentence be clearer by inserting “reproducible” before “live-cell imaging…”

Added

Line 56. Change “It becomes difficult…” to “These temporal steps make it difficult…”

Changed.

Line 57. Insert “better” before “understand”

Added.

Line 58. Replace “is critical” with “would be very useful”

Changed.

Line 66. Replace “angles” with “opportunities”

Changed.

Line 105. After “First,…” insert “in response to starvation…”

Added.

Line 109. Change “Second” to “In the second mechanism,…”

Changed.

Line 113. Replace “complete” with “occur”

Changed.

Line 118. Insert “particular” after “under”

Added.

Line 119. Replace “could also” with “are”

Changed.

Line 120. Replace “genuine” with “starvation-induced”

Changed.

Line 151. Change to “labels”

As FDAAs is the plural form of FDAA, we keep "label" unchanged. 

Lines 235-36. Sentence a bit awkward. How about “During development M. xanthus retains the MglA and MglB protein within their spores. Here, fluorescently-labeled MglB is seen to form …”

Changed following the reviewer's suggestion.

Line 247. Not clear what “inhibitors” refers to. Are they known?

We added ", such as mecillinam".

Line 253-54. Insert “are” before “always” and replace “longest” with “the farthest distance possible…”

Changed.

Line 254. Change “As the consequence” to “As a consequence”

Corrected. 

Line 270. Insert “move” after “complexes”

Changed to "which transport the Rod complexes toward the first pole".

Line 286: Insert “a” after “display”

Changed to "MreB filaments in M. xanthus display a rapid, directed motion that is not yet reported in other organisms".

Line 303: Replace “and the” with “, similar to how”

Changed to "Equipped with the Mgl regulators and gliding machineries, M. xanthus spores are able to regain fitness within one generation. "

Round 2

Reviewer 1 Report

It is difficult to read this review article. Because the article title differs from the content and the content is limited and fragmented. The authors describe Myxococcus xanthus rather than bacteria in general. I cannot understand why M. xanthus is a model organism for bacterial cell morphogenesis. The authors should describe M. xanthus more in detail in Introduction.

We added "As PG is essential for bacterial survival and its assembly systems are well conserved in most bacteria, including M. xanthus [23], M. xanthus provides unique opportunities to understand the mechanisms of PG assembly and rod-like morphogenesis. "

In this review paper, there is a significant lack of genetic information. Since it is mainly based on microscopic observation data, M. xanthus cannot be said to be a model organism for BACTERIAL cell morphogenesis, I think. The morphogenesis maintenance differs between Gram-positive and Gram-negative bacteria. In addition, mycoplasma are also a bacterium.

In the original article of PNAS, the authors used the words "phase 1" and "phase 2". But, in this review article, they used "spherical phase" and "vegetative growth". Does the spherical phase mean spore? If so, in this review article, morphological change means germination of spores. Thus, in the spherical phase, it is uncertain whether the spores can divide or not. Are spherical cells found in the vegetative growth? The authors explained the difference between spores of M. xanthus and endospores of Firmicutes in the section "The sporulation and germination of M. xanthus". I cannot understand why the "spores" of M. xanthus are recognized as spores. The authors should describe what are spores.

We believe that we have already explained the different phases in germination in our original manuscript. We wrote "These distinct growth patterns allow us to divide M. xanthus spore germination into two phases: the spherical phase (Phase I) in the first hour of germination when spores assemble PG evenly on their entire surfaces and the elongation phase (Phase II) in the second to third hour when PG growth at the nonpolar regions drives cell elongation (Fig. 1A). " To further distinguish spore and the spherical phase of germination, we also modified Fig. 1.

What is spore in this review paper? I cannot understand spores of M. xanthus that the authors say. In addition, what are different genetic regulations between different phases?

In the M. xanthus spore germination, Mgl proteins play an important role. The authors mentioned that Mgl of M. xanthus may related to Cdc42 of the eukaryote Schizosaccharomyces pombe. The explanation for Mgl and Cdc42 is inadequate and fragmented. The authors should explain the Mgl proteins more detail in this review article. For example, how Mgl proteins are distributed among Bacteria? Are Mgl proteins related to only Myxococcus? or spore forming bacteria? or Gram-negative bacteria?

We added "MglA-like GTPases distribute widely in phylogenically diverse bacteria. MglA and its eukaryotic homologs are proposed to have evolved from a common ancestor [96]. "

The authors did not show the relation between MglA and Cdc42. What are eukaryotic homologs of MglA?

The authors did not show the cell cycle of M. xanthus. They showed only gemination of M. xanthus spores. In order to show that M. xanthus is a model bacterium for bacterial cell morphogenesis. The authors should show the cell cycle and explain when DNA is replicated, and cell wall and membrane are synthesized.

We clarified that "Vegetative M. xanthus cells are long rods (1 mm in diameter and 5 – 10 mm in length) that go through cell cycles similar to other rod-shaped model organisms, such as E. coli and B. subtilis [19,20]."

The authors did not show the cell cycle of M. xanthus.